

# Effects of afforestation with *Pinus sylvestris* var. *mongolica* plantations combined with enclosure management on soil microbial community

Jiaojiao Deng[1,2], Yongbin Zhou[1,2], Wenxu Zhu[1,2] and You Yin[1,2]

[1] College of Forestry, Shenyang Agricultural University, Shenyang, Liaoning, China
[2] Research Station of Liaohe-River Plain Forest Ecosystem, Chinese Forest Ecosystem Research Network (CFERN), Shenyang Agricultural University, Tieling, Liaoning, China

Corresponding author
You Yin, 1993500012@syau.edu.cn

## ABSTRACT

Grazing and litter removal can alter understory structure and composition after afforestation, posing a serious threat to sustainable forest development. Enclosure is considered to be an effective measure to restore degraded forest restoration. However, little is known about the dynamics of soil nutrients and microbial communities during the forest restoration process. In the present study, the effects of *Arachis hypogaea* (AH), *Pinus sylvestris* var. *mongolica* (PSM) and *Pinus sylvestris* var. *mongolica* with enclosure (PSME) on soil chemical properties and soil microbial communities were studied in Zhanggutai, Liaoning Province, China. The results showed that PSME could remarkably contribute to improve soil total C, total N and total P compared to PSM and AH. Additionally, PSM could clearly increase the soil bacterial community diversity and fungal Chao1 index and ACE index. Additionally, PSME could further increase soil Chao1 index and ACE index of soil bacteria. Soil total C, total N and available N were the main factors related to soil microbial diversity. Actinobacteria and Ascomycota were the predominant bacterial and fungal phyla, respectively. Specifically, PSME could increase the relative abundances of Actinobacteria, Gemmatimonadetes, Ascomycota and Mortierellomycota and decreased the relative abundances of Acidobacteria, Chloroflexi and Basidiomycota than PSM. PSM and PSME could clearly change soil microbial communities compared with AH and PSME could remarkably shift soil fungal communities than PSM. What's more, the soil microbial community structure were affected by multiple edaphic chemical parameters. It can be seen that afforestation combined with enclosed management potentially regulate microbial properties through shifting the soil properties. This study can provide new ideas for further understanding the impact of enclosure on PSM and provide theoretical support for the management of PSM.

# INTRODUCTION

The arid and semi-arid regions cover an area of about 4 billion hectares, accounting for about 30% of the terrestrial land area within the globe (*Lal, 2001*). These areas have been

continuously threatened by soil degradation and desertification due to natural and human interference such as climate variation, over-cultivation intensity, overgrazing of livestock for decades, vegetation destroyed by firewood (*He et al., 2015*; *Haberl et al., 2007*; *Li et al., 2018*), which may lead to a range of severely ecological problems, such as loss of biodiversity, soil erosion, global soil C loss (*Lal, 2001*). It has been estimated that land degradation caused by desertification affects one-quarter of the world's land surface, containing one-fifth of the world's population, mostly living in developing countries (*D'Odorico et al., 2013*). In view of this, a series of ecological restoration programs have been implemented, among which, re-vegetation through afforestation is one of the most commonly used techniques (*Nunezmir et al., 2015*) and an effective method to combat desertification (*Gao et al., 2002*), and increase soil C and N storages (*Su et al., 2005*; *Hu et al., 2008*; *Chen et al., 2010*), especially in arid and semi-arid regions.

While, the effects of forest management and disturbance can be extended to change the structure and function of forest ecosystems over time (*Fox, 2000*). For a long time, it has been believed that forests around the world principally have been threatened by logging for fuel or industrial use and land reclamation of pasture and agriculture (*Lindquist et al., 2012*). Nevertheless, in some regions, litter removal and livestock grazing could endanger long-term protection of forest (*Fleischner, 1994*; *Belsky & Blumenthal, 1997*). For example, throughout the 19th Century, the collection of litter from managed forests as fuel and farming have been widely practiced in many countries, still existing in some regions (*Hofmeister et al., 2008*; *Chevasco et al., 2016*), which may have negative effect on nutrient cycling and forest productivity. In addition, the use of forests by livestock could have substantial impacts on the structure and dynamics of forests, soil microenvironment, land vegetation cover, water availability, plant establishment (*Yates, Norton & Hobbs, 2000*), soil quality (*Li et al., 2012a*), wildlife conservation and the other ecosystem service functions (*Vargas et al., 2000*). In view of this, a series of policies of ecological conservation have been launched for the sake of reducing its negative impact on forests (*Mu et al., 2013*).

Enclosure fencing is identified as an effective management practice to enhance the carbon sequestration potential of ecosystems and restore degraded ecosystems (*Yao et al., 2018*), which has been extensively concerned and used by countries all over the world (*Leigh & Holgate, 1979*; *Gebregergs et al., 2019*). For example, several forest enclosure trials have reported that forest enclosure could increase the diversity and density of saplings and seedlings, quantity of natural herb, understory vegetation, as well as the cover of litters (*Cabin et al., 2000*; *Miller & Wells, 2003*; *Dodd & Power, 2007*), and decrease soil compaction and erosion (*Spooner, Lunt & Robinson, 2002*; *Michels, Vieira & De Sá, 2012*), subsequently increasing the soil organic C and N derived from litter decomposition and root exudation (*Bai et al., 2012*; *Wu et al., 2014*), facilitating forest establishment (*Speed et al., 2014*). Similarly, the results in the hillside forests of Uruguay indicated that exclusion of livestock with fence from the forest preferably boost soil conditions and provide opportunities for the regeneration of certain species (*Etchebarne & Brazeiro, 2016*). However, other experiments found that fenced forests did not reveal evidence of tree regeneration or soil structure improvement (*Fischer et al., 2009*; *Sankey, 2012*). Therefore,
the effect of fences excluding livestock on forest are complexity and variability, and there are lacking adequate evidences, particular in terms of the impact of enclosures on microorganisms.

Soil microorganisms play pivotal roles in the decomposition of tree litter, and nutrient mineralization (*Xu et al., 2008*; *Burton et al., 2010*). Compared to soil physic-chemical properties and changes in aboveground vegetation, microbes are more dynamic and sensitive to any small variations in soil or environmental stress (*Cruz-Paredes et al., 2017*), which can generally be considered as early indicators of the human induced effects on soil condition changes and soil ecology (*Honeker et al., 2017*; *Vinhal-Freitas et al., 2017*; *Yang et al., 2018*). So far, intensive studies have concerned the effects of enclosure on soil on soil physical, chemical and biological properties (*Yong-Zhong et al., 2005*), as well as soil nematode community (*Zhang et al., 2019*). However, researches on the effects of afforestation with enclosure management measures on soil microorganisms are insufficient.

In China, the current "Three North" Shelterbelt Development Program, started in 1978, is the largest afforestation project in China and even worldwide (*Li et al., 2012b*). *Pinus sylvestris* var. *mongolica* (PSM) naturally distributed in Honghuaerji, as the main afforestation tree species, was first introduced to Zhanggutai, Liaoning Province, China (*Zhou et al., 2019*), which has been successfully promoted in more than 300 counties in 13 provinces and regions. According to the results of the eighth forest resource inventory (2009–2013), the area of PSM plantation has reached $4.17 \times 10^5$ ha, which has made important contribution to ecosystem service functions. However, due to the extensive grazing and litter removal in the PSM plantations, the surface loses the litter layer that maintains the moisture and ecosystem material circulation, resulting in severe soil degradation (*Willcock et al., 2016*), which has become a bottleneck and difficult problem to constraint the sustainable development of plantations. About the enclosure of PSM, the study from *Zhang et al. (2012)* suggested that the thickness and weight of litters of PSM in Zhanggutai area at the southern margin of Horqin sandy land increased significantly with the increase of the enclosure years. However, none of studies measured, in detail, particularly the effect of afforestation with *Pinus sylvestris* var. *mongolica* plantations combined with enclosure management (PSME) on soil microbial communities.
In addition, the vegetation type was *Arachis hypogaea* (AH) (peanut) farmland prior to afforestation, and AH were selected as the control. Thus, in present study, we investigated the responses of soil characteristics and microbial communities to afforestation with PSME management in Zhanggutai using the High-throughput sequencing technology. On the basis of former studies, we put three assumptions (i) PSM and PSME contribute to better increase soil nutrient accumulation compared to AH, especially PSME; (ii) PSM and PSME could clearly shift soil microbial communities compared with AH; (iii) furthermore, soil microbial communities existed clear differences between PSME and PSM; (iv) PSM and PSME potentially regulate microbial properties through shifting the soil properties. This research could provide a new insight into the effects of enclosure on *Pinus sylvestris* and supply theoretical support for enclosure management of *Pinus sylvestris*.

## MATERIALS AND METHODS

### Site information

The study area was located in Zhanggutai (42°37′30″–42°50′00″N and 122°11′15″–122°30′00″E), north of Zhangwu county, northwest edge of Liaoning Province, China, which belongs to the north temperate continental monsoon climate with four distinct seasons, rain and heat in the same season, sufficient sunshine, large temperature difference between day and night. The annual average temperature is 5.7 °C with the highest of 35.2 °C and the lowest of −29.5 °C. The annual precipitation is 450–550 mm, and annual evaporation of 1,200–1,450 mm. The average elevation is 226.5 m. The strong wind frequently occurred in spring and winter, and the instantaneous maximum wind speed is 32 m/s. And the sandstorms blows up to more than 240 times with speed of more than 5 m/s per year. The average frost-free period was 156 days. Soil type is classified as Cambic Arenosols of sandy origin (*IUSS Working Group WRB, 2007*) with characteristic of coarse texture and loose structure (*Li et al., 2012a*).

### Determination of vegetation type and soil samples collection

Four 1 ha sites of PSM plantations planted in 1986 with same site conditions in Zhanggutai (42°40′01″N, 122°29′55″E; 42°39′49″N, 122°30′00″E; 42°40′02″N, 122°29′40″E; 42°42′39″N, 122°28′49″E) were selected. The extensive grazing and litter removal in the PSM plantations were always existed. In order to promote the sustainable development of PSM plantations, one permanent enclosure plot with 50 × 50 m was established in each site in July, 2009. The same degree of disturbance existed outside the enclosure in four sites, and plot of 50 × 50 m was randomly built up in each disturbance site. Prior to afforestation, the vegetation type was AH (peanut) farmland, thus, we selected adjacent AH with 50 × 50 m as the control. After removing the litter layer, soil samples were collected from 8 to 12 core points for each plot with using a soil auger of 2.5 cm in diameter and 0–10 cm of the depth and mixed as a composite soil sample, giving a total 12 soil samples. All soil samples were placed in an ice box, taken back to the laboratory, and roots and other debris were removed and discarded. These soil samples were divided into two parts, and one part for the determination of soil microbial communities was sieved through a 2-mm screen and stored −80 °C immediately. While, another part for the determination of soil characteristics was air-dried at room temperature. Simultaneously the measurements of vegetation were completed in July 2019 (Table 1). In each plot, the diameter at breast height and tree height were measured using a breast diameter ruler and a clinometer for all trees, respectively. Five 1 × 1 m$^2$ quadrat were randomly established in each plot and sampled for both accumulated litter and understory plant biomass.

### The determination of soil properties

The soil pH was determined in a 1:2.5 soil-water suspension using a pH meter (MT-5000; Sanbon, Shanghai, China) with a glass electrode. The concentrations of total carbon (C) and total nitrogen (N) were assessed via an elemental analyzer (EA3000; Euro Vector, Pavia, Italy) with air-dried soil passed through a 0.2-mm screen. Total phosphorus (P)

**Table 1 Site characteristics.**

| Different samples | Plant age (years) | Stand density (plant·hm⁻²) | Trees | | | Understory vegetation | | | Surface litter (cm) |
|---|---|---|---|---|---|---|---|---|---|
| | | | Diameter at beast height (cm) | Tree height (m) | Canopy density (%) | Coverage (%) | Aboveground biomass (g·m⁻²) | Number species | |
| PSM | 35 | 495 | 21.63 | 11.55 | 35 | 90 | 66.02 | 10 | 1 |
| PSME | 35 | 573 (48 regeneration seedlings) | 24.33 | 12.5 | 37 | 45 | 120.83 | 25 | 5 |

**Note:**

Measurements of the diameter at beast heigh and tree height did not include regeneration seedlings. PSM, *Pinus sylvestris* var. *mongolica*; PSME, *Pinus sylvestris* var. *mongolica* with enclosure.

concentration was estimated with sulfuric acid-soluble perchlorate acid-molybdenum antimony colorimetric method. The alkali-hydrolysis and diffusion method was used to measure soil available N content. And the available P was extracted by $NaHCO_3$ (0.5 mol·L⁻¹) and measured using the antimony molybdenum anti-colorimetric method.

## Soil DNA extraction and amplification sequencing

The DNA of samples was extracted from 0.5 g of soil with the FastDNA SPIN Kit for Soil (MP Biomedicals, Santa Ana, CA, USA), according to the manufacturer's instructions. The genomic DNA was amplified by PCR using the 16 S rRNA gene V3–V4 hypervariable regions region primers (338F and 806 R) (*Xu et al., 2016*), and the Internal Transcribed Spacer (ITS) gene regions primers (ITS1F-ITS2) (*Caban et al., 2018*; *Nottingham et al., 2018*). PCR reactions: 8.75 µl of $ddH_2O$; 5 µl of Q5 reaction buffer (5×) and Q5 High-Fidelity GC buffer (5×), respectively; 2 µl of dNTPs (2.5 mM) and DNA Template, respectively; 1 µl (10 µM) of forward primer and reverse primer, respectively; 0.25 µl (5 U/µl) of Q5 High-Fidelity DNA Polymerase, a total of 25 µl mixture. PCR thermal cycling condition: an initial denaturation step at 98 °C for 5 min, then 25 cycles (denaturation at 98 °C for 15 s, annealing at 55 °C for 30 s and elongation at 72 °C for 30 s), with final elongation step of 72 °C for 5 min. Agencourt AMPure Beads (Beckman Coulter, Indianapolis, IN, USA) and PicoGreen dsDNA Assay Kit (Invitrogen, Carlsbad, CA, USA) were used to purify and quantify PCR amplicons. Finally, the Illumina's MiSeq PE300 platform (Shanghai Personal Biotechnology Co., Ltd., Shanghai, China) was used for sequencing. The raw high-throughput sequencing data of bacteria and fungi were stored in the NCBI database with the accession number SRA accession of PRJNA562091 and PRJNA562096, respectively.

## Statistical analyses

The differences of soil characteristics and microbial community diversity under different treatments were performed as means ($n = 4$) ± standard errors and analyzed using one-way ANOVA with the least-significant-difference test. The relationships between soil microbial diversity and chemical properties were explored using Spearman's rank correlation and visualized using R with the package of "corrplot". The shared and unique OTUs among different treatments were calculated and visualized using the

**Table 2 Soil chemical properties under different treatments.**

| Different samples | pH value | Total C/g·kg⁻¹ | Total N/g·kg⁻¹ | C/N ratio | Total P/g·kg⁻¹ | Available P/mg·kg⁻¹ | Available N/mg·kg⁻¹ |
|---|---|---|---|---|---|---|---|
| AH | 5.68 (0.02)bB | 3.96 (0.46)cC | 0.57 (0.05)cC | 7.00 (0.29)cC | 0.169 (0.009)aAB | 27.25 (2.56)aA | 35.39 (2.89)bA |
| PSM | 5.95 (0.10)aA | 6.70 (0.69)bB | 0.73 (0.07)bB | 9.21 (0.31)bB | 0.143 (0.015)bB | 17.27 (6.90)bAB | 43.35 (4.45)bA |
| PSME | 5.61 (0.05)bB | 8.98 (0.50)aA | 0.91 (0.06)aA | 9.90 (0.19)aA | 0.173 (0.010)aA | 15.69 (4.40)bB | 46.00 (7.08)aA |
| *F* test | 31.68 | 80.45 | 31.22 | 128.33 | 7.59 | 6.42 | 4.68 |
| *P* value | 0.0001 | 0.0001 | 0.0001 | 0.0001 | 0.01 | 0.02 | 0.04 |

Note:
Data are means (standard error) (*n* = 4). AH, *Arachis hypogaea*; PSM, *Pinus sylvestris* var. *mongolica*; PSME, *Pinus sylvestris* var. *mongolica* with enclosure. Different uppercase letters in the same column indicate significant differences at the 0.01 level, and different lowercase letters indicate significant differences at the 0.05 level.

packages of "venndiagram" in R. Non-metric multidimensional scaling (NMDS) analysis based on unweighted uniFrac and weighted unifrac distance matrix was used to investigate the distribution characteristics and the dissimilarity of soil bacterial and fungal communities among sites using R with the package of "vegan", respectively. Based on Bray–Curtis distance matrix, heatmap plots of soil bacteria and fungi with the relative abundances of top 50 were performed using R with the package of "vegan". Redundancy analysis (RDA) was further conducted to determine the relationships between soil chemical properties and soil microbial community structure using R with the packages of "vegan" and "car". Variance Partitioning Analysis (VPA) was used to quantitatively assess the contribution of each environmental factor to the microbial communities using R with with the package of "vegan". Associated network analysis was used to calculate the correlation between soil characteristics and soil dominant bacterial and fungal phyla and visualized in Gephi.

## RESULTS

### Soil properties among different treatments

Soil pH in this region exhibited acid soil, and the maximum value occurred in PSM with 5.95, and distinctly higher than that of AH and PSME ($P < 0.01$) (Table 2). Significant differences were observed in total C ($F = 80.45$, $P < 0.00$), total N ($F = 31.22$, $P < 0.00$), total P ($F = 7.59$, $P = 0.01$), available P ($F = 6.42$, $P = 0.02$) and available N ($F = 4.68$, $P = 0.04$) among AH, PSM and PSME. The values of soil total C, total N, and available N in AH exhibited the lowest with 3.96 g·kg⁻¹, 0.57 g·kg⁻¹ and 35.39 mg·kg⁻¹, respectively. Site PSME could remarkably increase soil total C, total N, total P, which were 1.34, 1.25 and 1.21 folds in compared to site PSM. Similarly, the available N concentration in PSME was 1.06 times than this of PSM. While, the value of soil available P were 20.9% higher in site PSM relative to the PSME land (Table 2).

### Soil microbial community diversity among different treatments

A total of 582,908 bacterial valid sequences and 608,220 fungal effective sequences were acquired, with an average of 48,575 and 50,685, respectively, which were clustered into 6,443 OTUs and 1,257 OTUs according to the 97% similarity threshold. Rarefaction curves based on 97% similarity level tended to be flat with the increase of 16S rDNA and ITS rDNA sequences, suggesting that the number of sequences was enough and reasonable

(Fig. S1). Bacterial OTUs identified in AH, PSM and PSME were 3,663, 4,821 and 4,603, with shared OTUs of 2,176 and unique OTUs of 753, 677 and 545, respectively (Fig. 1A). Fungal OTUs identified in AH, PSM and PSME were 502, 736 and 783, with shared OTUs of 166 and unique OTUs of 225, 197 and 237, respectively (Fig. 1B). The NMDS plot of bacterial OTUs indicated that soil bacterial communities among AH, PSM and PSME existed significant difference (stress = 0.02), and microbial community assemblage in the AH remarkably differed compared with the PSM and PSME (Fig. 2A). For PSM and PSME, the soil bacterial communities existed some overlapping, manifesting that the soil bacterial community were similar (Fig. 2A). Similar patterns were found in soil fungal communities (stress = 0.01), however, soil fungal communities from PSM and PSME exhibited no overlapping (Fig. 2B). Thus it can be seen afforestation with PSM could change soil microbial communities, and the effects of PSME on soil fungal communities were significantly greater than those of bacterial communities (Fig. 2).

Indicators of bacterial community diversity, including Simpson index and Shannon index, displayed no obvious differences among AH, PSM and PSME. Similar patterns were observed in the fungal Simpson index and Shannon index (Table 3). The bacterial Chao1 index and ACE index in PSME were the highest with 3,061.60 and 3,082.46, respectively, followed by PSM and AH, no significant differences compared to PSM ($P > 0.05$). Additionally, fungal Chao 1 index and ACE index showed a similar trend with the maximum values in PSME of 413.35 and 419.45, respectively (Table 3). The relationships between soil microbial community diversity and soil characteristics were calculated with Spearman's rank correlation (Fig. 3). Soil bacterial Chao1 index and ACE index showed significantly positive correlations with total C, total N, available N and C/N (Fig. 3A). With regard to soil fungal community diversity, soil fungal Chao1 index and ACE index had significantly positive relations with soil total C, total N and available N (Fig. 3B). Accordingly, soil total C, total N and available N were the main factors effecting the soil microbial diversity (Fig. 3).

## Soil microbial community composition among different samples

In case of soil bacteria, the dominant bacterial communities with the relative abundances great than 1% were Actinobacteria (35.42–43.37%), Proteobacteria (21.95–25.84%), Acidobacteria (5.61–19.99%), Chloroflexi (7.68–12.55%), WPS-2 (1.77–4.13%), Gemmatimonadetes (2.00–3.09%), Patescibacteria (2.03–2.55%), Planctomycetes (2.02–2.37%), Bacteroidetes (1.44–2.06%) and Firmicutes (0.78–1.35%), accounting for 97.74%, 97.38% and 98.30% in AH, PSM and PSME (Fig. 4A). No significant differences were observed in Proteobacteria, WPS-2, Patescibacteria, Planctomycetes, Bacteroidetes and Firmicutes ($P > 0.05$). Significant changes in the relative abundances of Actinobacteria ($F = 13.146$, $P = 0.0021$), Acidobacteria ($F = 87.849$, $P = 0.0001$) and Gemmatimonadetes ($F = 9.885$, $P = 0.0054$) were observed with the different samples. Specifically, PSM obviously increased the relative abundance of Acidobacteria, and decreased the relative abundance of Actinobacteria compared to AH ($P < 0.05$). We further found that PSME could increase the Actinobacteria and Gemmatimonadetes with the relative abundance of 43.37% and 2.33%, while, which decreased the Acidobacteria and Chloroflexi with 14.90%

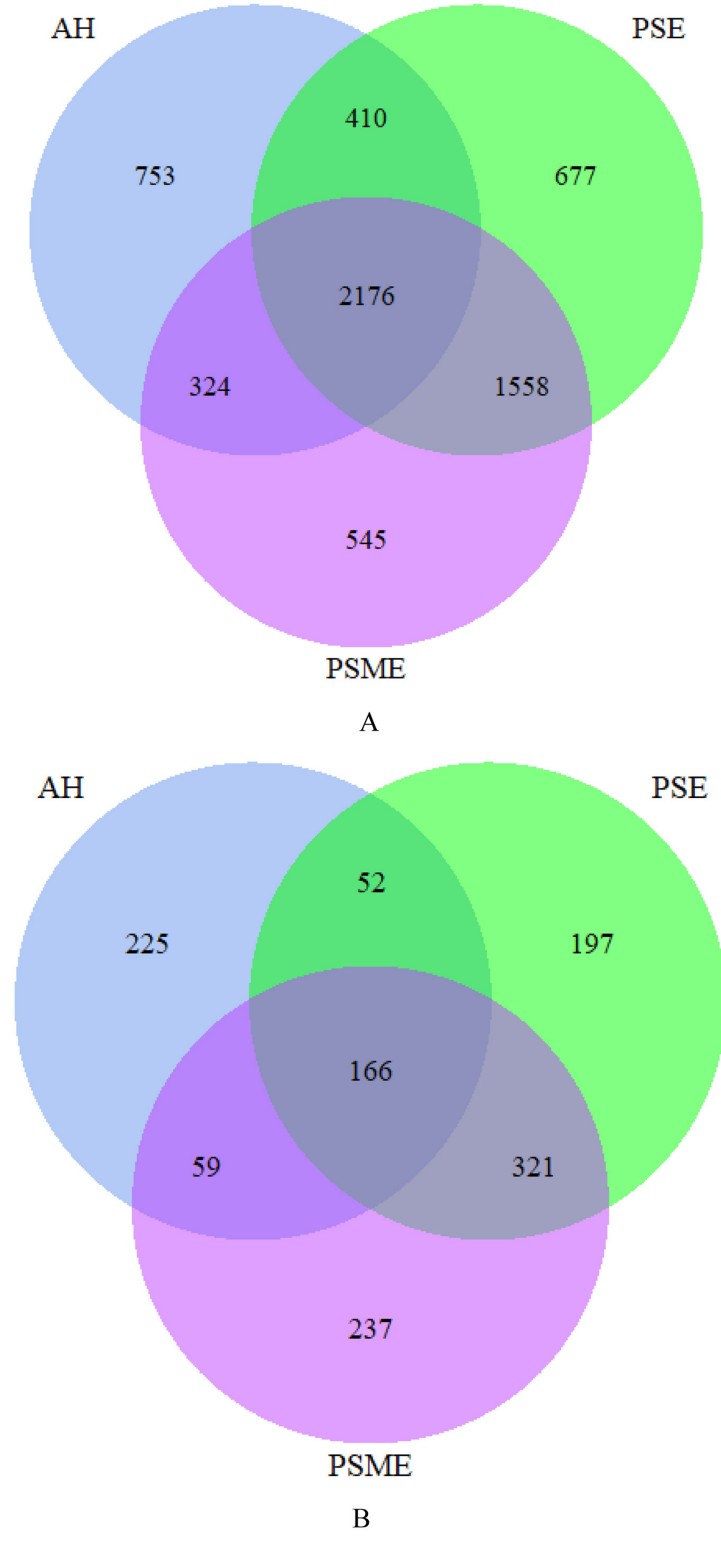

**Figure 1 Venn diagrams of the shared and unique bacterial OTUs (A) and fungal OTUs (B).** AH, *Arachis hypogaea*; PSM, *Pinus sylvestris* var. *mongolica*; PSME, *Pinus sylvestris* var. *mongolica* with enclosure.

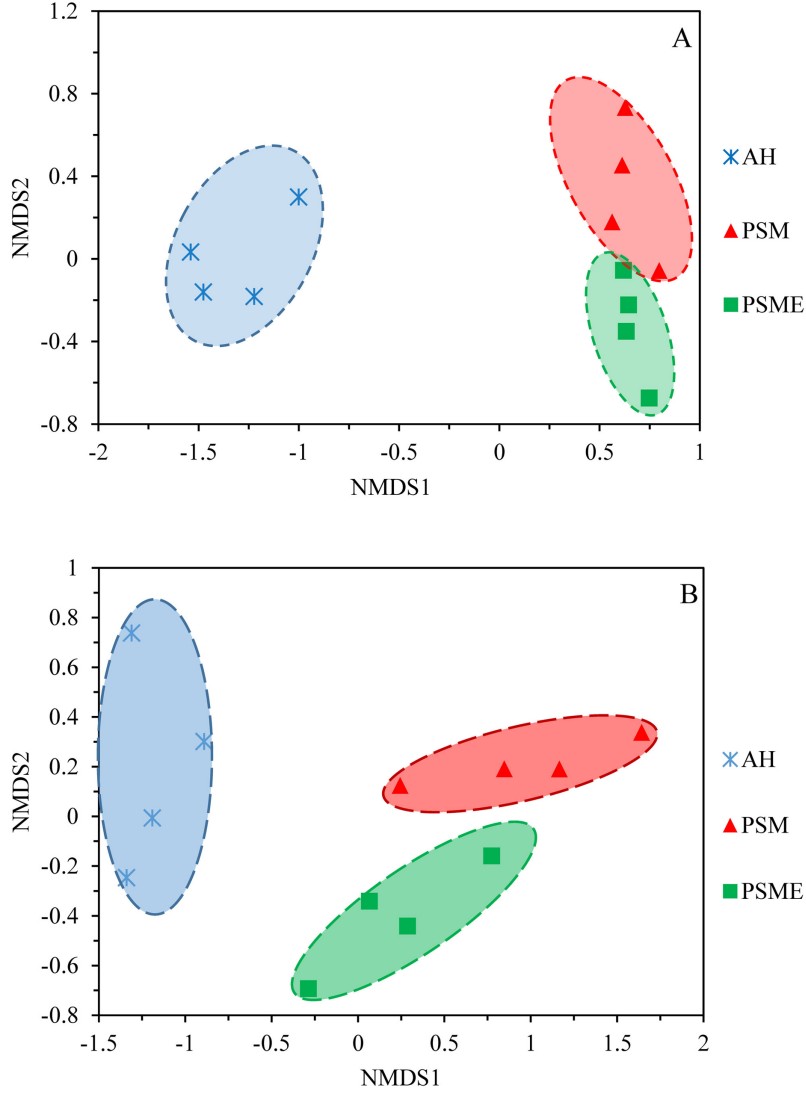

**Figure 2 NMDS plots of bacterial OTUs (A) and fungal OTUs (B) based on unweighted and weighted unifrac, respectively.** AH, *Arachis hypogaea*; PSM, *Pinus sylvestris* var. *mongolica*; PSME, *Pinus sylvestris* var. *mongolica* with enclosure.

and 7.68% compared to PSM ($P < 0.05$) (Fig. 4A). With regard to soil fungi, the dominant fungal communities were Ascomycota, Basidiomycota and Mortierellomycota, accounting for 86.49%, 97.47% and 97.67% in AH, PSM and PSME (Fig. 4B). Compared to AH, PSM dramatically decreased Ascomycota and increased Basidiomycota ($P < 0.05$). PSME could significantly increase Ascomycota, Mortierellomycota, and decrease the relative abundance of Basidiomycota ($P < 0.05$) (Fig. 4B).

At the genus level, soil bacterial genera with the relative abundances greater than 1% were *Crossiella*, *Jatrophihabitans*, *Conexibacter*, *Sphingomonas*, *Blastococcus*, *Acinetobacter*, *Bradyrhizobium*, *RB41*, *Mycobacterium*, *Candidatus-Solibacter*, *Nocardioides*, *Gemmatimonas* and *Bryobacter* (Fig. 5A) among AH, PSM and PSME. PSME site improved the relative abundance of *Crossiella*, and decreased the relative

**Table 3 Changes in soil microbial diversity under different treatments.**

| Different samples | Simpson index | | Chao1 index | | ACE index | | Shannon index | |
|---|---|---|---|---|---|---|---|---|
| | Bacteria | Fungi | Bacteria | Fungi | Bacteria | Fungi | Bacteria | Fungi |
| AH | 0.996 ± 0.001aA | 0.94 ± 0.01aA | 2,111.71 ± 110.85bB | 268.04 ± 20.26bB | 2,126.62 ± 119.64bB | 271.91 ± 19.05bB | 9.58 ± 0.24bA | 5.44 ± 0.24aA |
| PSM | 0.996 ± 0.001aA | 0.88 ± 0.06bA | 3,003.22 ± 299.64aA | 388.74 ± 96.05aAB | 3,047.36 ± 354.48aA | 391.94 ± 95.61aAB | 10.05 ± 0.22aA | 5.08 ± 1.02aA |
| PSME | 0.993 ± 0.002aA | 0.95 ± 0.03aA | 3,061.60 ± 410.84aA | 413.35 ± 30.87aA | 3,082.46 ± 397.57aA | 419.45 ± 30.46aA | 9.87 ± 0.37abA | 6.05 ± 0.71aA |
| F test | 2.58 | 4.10 | 12.56 | 6.84 | 11.83 | 7.08 | 2.83 | 1.80 |
| P value | 0.13 | 0.054 | 0.002 | 0.02 | 0.003 | 0.01 | 0.11 | 0.22 |

**Note:**
Data are means ± standard error (n = 4). AH, *Arachis hypogaea*; PSM, *Pinus sylvestris* var. *mongolica*; PSME, *Pinus sylvestris* var. *mongolica* with enclosure. Different uppercase letters in the same column indicate significant differences at the 0.01 level, and different lowercase letters indicate significant differences at the 0.05 level.

abundance of *Sphingomonas* compared to PSM (Fig. 5A). The soil fungal genera with relative abundances greater than 1% were *Penicillium*, *Mortierella*, *Amphinema*, *Trechispora*, *Pseudogymnoascus*, *Lectera*, *Didymella*, *Wilcoxina*, *Papiliotrema* and *Talaromyces* among AH, PSM and PSME (Fig. 5B). *Amphinema* and *Wilcoxina* were only existed in PSM and PSME, which are very common ectomycorrhizal fungi (Fig. 5B). On this view, afforestation with *Pinus sylvestris* could increase the relative abundances of soil ectomycorrhizal fungi. What's more, PSME site improved the relative abundances of *Penicillium*, *Mortierella*, *Amphinema*, *Trechispora* and *Pseudogymnoascus*, and reduced the relative abundances of *Trechispora* and *Wilcoxina* relative to PSM (Fig. 5B).

The heatmap plots of soil bacterial and fungal communities among different samples were divided into two clusters, including AH and PSM plus PSME, respectively, indicating that afforestation with PSM might remarkably shift soil microbial communities and the microbial communities between PSM and PSME were similar but also clearly different (Fig. 6).

## The correlations between soil microbial community composition and soil environment factors

Redundancy analysis between soil bacterial OTUs great than 1 and soil characteristics indicated that the first two principal components explained approximately 54.84% and 25.22% of the total variability, respectively (Fig. 7A). Soil total C ($r = 0.92$), total N ($r = 0.87$), C/N ($r = 0.91$), available P ($r = -0.72$) and available N ($r = -0.68$) had lager contributions to RDA1. While, soil pH ($r = 0.83$) and total P ($r = -0.82$) showed great relations with RDA2. The proportion of explanation of each soil constraint factors to microbial community were analyzed based on VPA. Soil pH, total C, total N, C/N, total P, available P and available N alone explained 2.60%, 4.39%, 4.74%, 2.50%, 7.95%, 4.64% and 3.10% of the total variables (Table 4). The bacterial communities in AH formed individual cluster, however, bacterial communities from PSM and PSME existed some overlapping, indicating that soil bacterial community between PSM and PSME were more similar (Fig. 7A). Overtly, the bacterial communities of AH were strongly related to higher

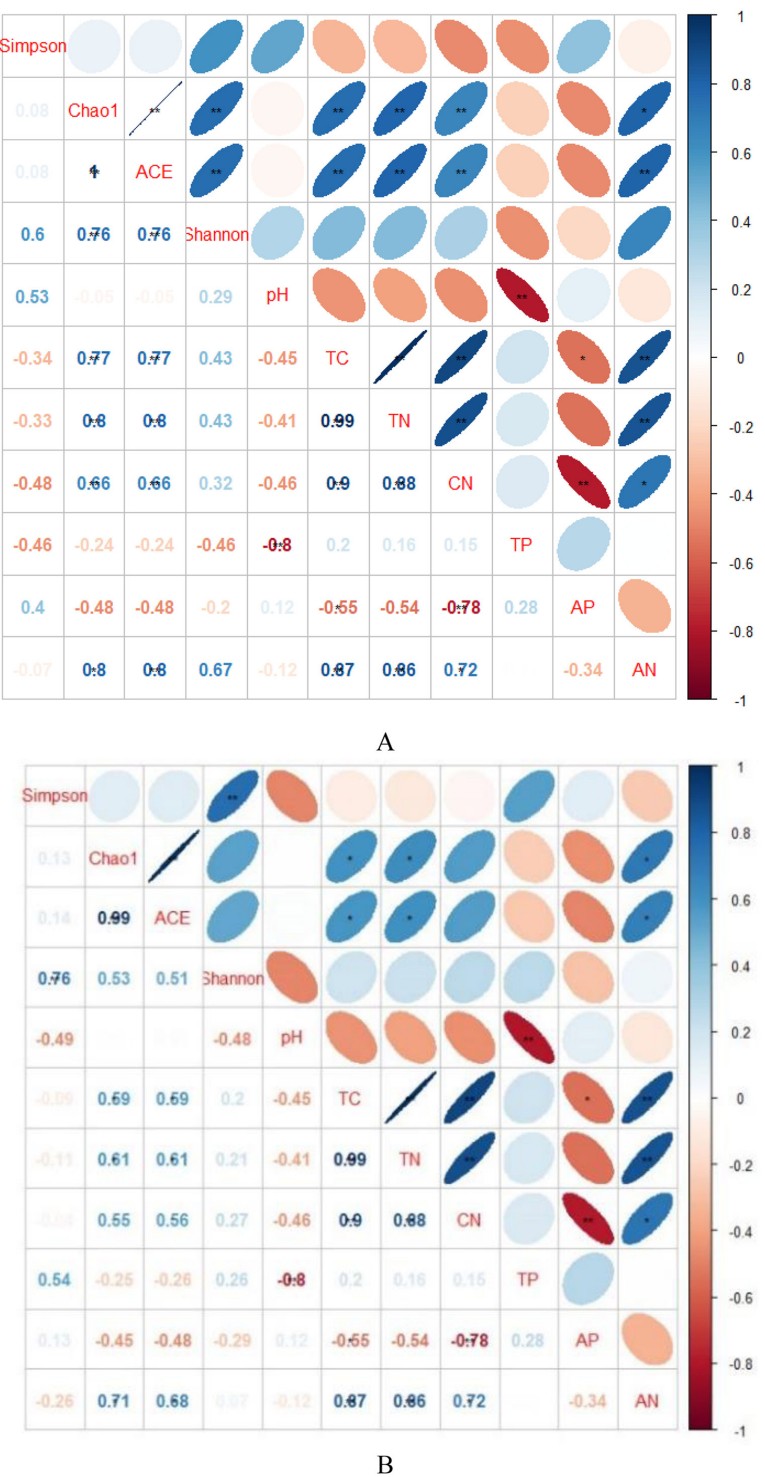

**Figure 3 Spearman's rank correlations between soil characteristics and soil bacterial (A) and fungal diversity indices (B).** TC, total C; TN, total N; TP, total P; AP, available P; AN, available N; CN, C/N. Blue indicates that the two variables are positively correlated, while red indicates that the variables are negatively correlated. The flatter the ellipse indicates that the absolute value of the correlation coefficient is larger; the rounder the ellipse indicates that the absolute value of the correlation coefficient is smaller.

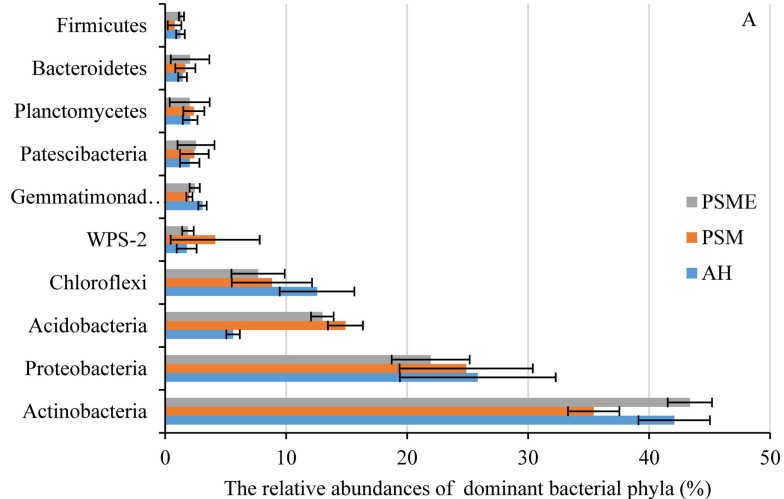

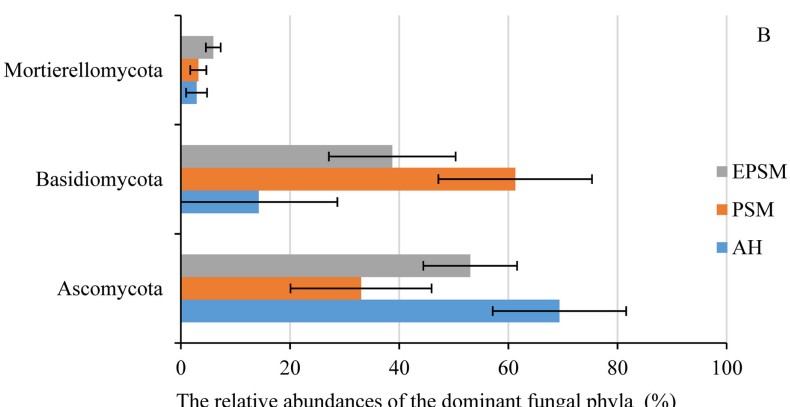

**Figure 4 The relative abundances of dominant bacterial (A) and fungal (B) phyla among different samples.** AH, *Arachis hypogaea*; PSM, *Pinus sylvestris* var. *mongolica*; PSME, *Pinus sylvestris* var. *mongolica* with enclosure.      

soil available P. In contrast, the contents of soil total C, total N, available N and C/N did not correlate with the single cluster of AH (Fig. 7A).

In case of soil fungi, the CCA1 and CCA2 captured 34.91% and 16.99% of the total variations, and revealed that the fungal OTUs were divided into three distinctly different cluster groups, including AH, PSM and PSME, respectively (Fig. 7B). Furthermore, soil total C ($r = 0.86$), total N ($r = 0.79$), C/N ($r = 0.95$), available P ($r = -0.75$) and available N ($r = -0.71$) made a great contribution to CCA1. And soil pH ($r = -0.79$) and total P ($r = -0.67$) had a large proportion to the CCA2 (Fig. 7B). Apparently, the fungal communities of AH were intensively linked to higher soil available P, and the fungal communities of PSM were strongly related to higher soil pH value (Fig. 7B). Furthermore, soil pH, total C, total N, C/N, total P, available P and available N alone explained 9.20%, 7.94%, 7.81%, 8.21%, 7.33%, 6.56% and 5.56% of the total variables (Table 4).

In terms of soil dominant bacterial and fungal phyla, we observed that the relative abundance of Actinobacteria existed significantly negative correlation with soil pH

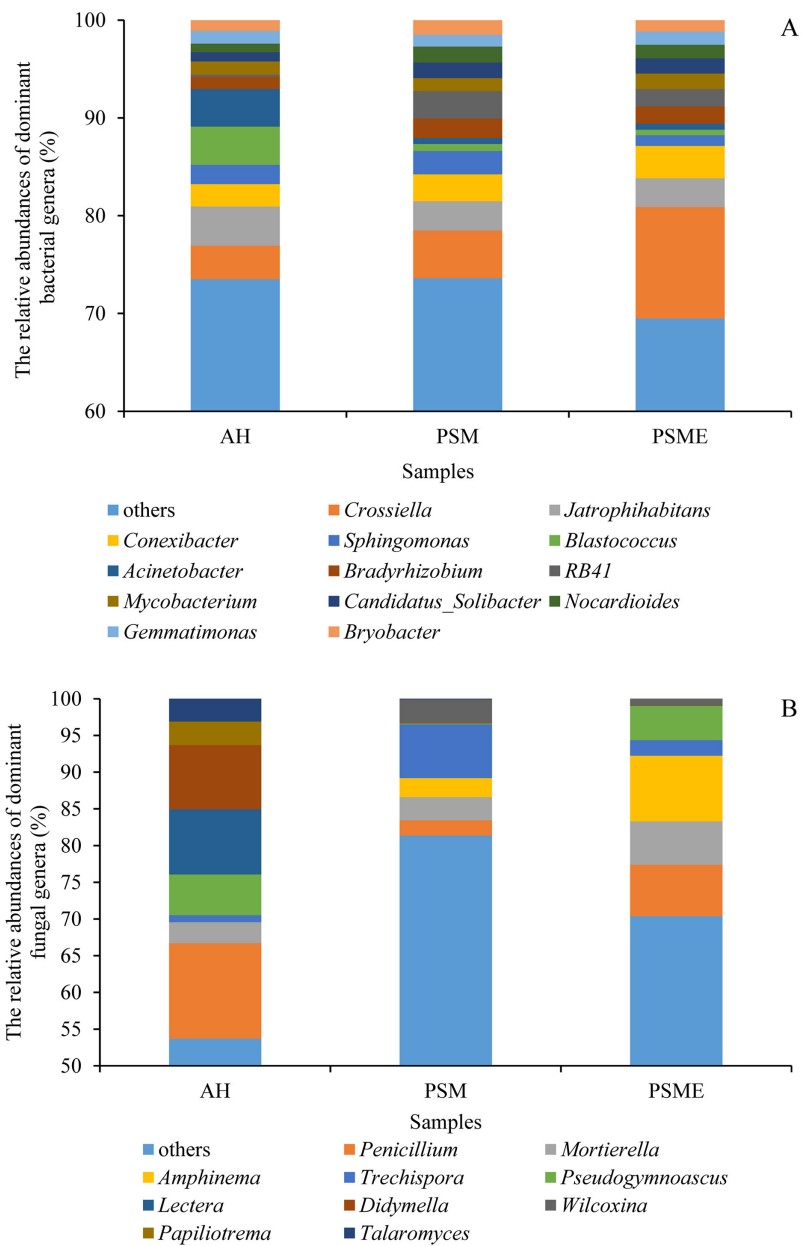

**Figure 5 The relative abundances of dominant bacterial (A) and fungal (B) genera among different samples.** AH, *Arachis hypogaea*; PSM, *Pinus sylvestris* var. *mongolica*; PSME, *Pinus sylvestris* var. *mongolica* with enclosure.

($r = -0.73$, $P = 0.01$). While, the relative abundance of Actinobacteria dramatically increased with the increase of soil total P content ($r = 0.82$, $P = 0.01$). The relative abundance of Chloroflexi had significantly negative correlation with C/N ($r = -0.78$, $P = 0.01$), while, which had dramatically positive relation with soil total P ($r = 0.90$, $P = 0.01$). The relative abundance of Acidobacteria increased with the increase of soil available N ($r = 0.79$, $P = 0.01$) and C/N ($r = 0.59$, $P = 0.05$). The relative abundance of Gemmatimonadetes increased with the increase of total P ($r = 0.64$, $P = 0.05$) and available
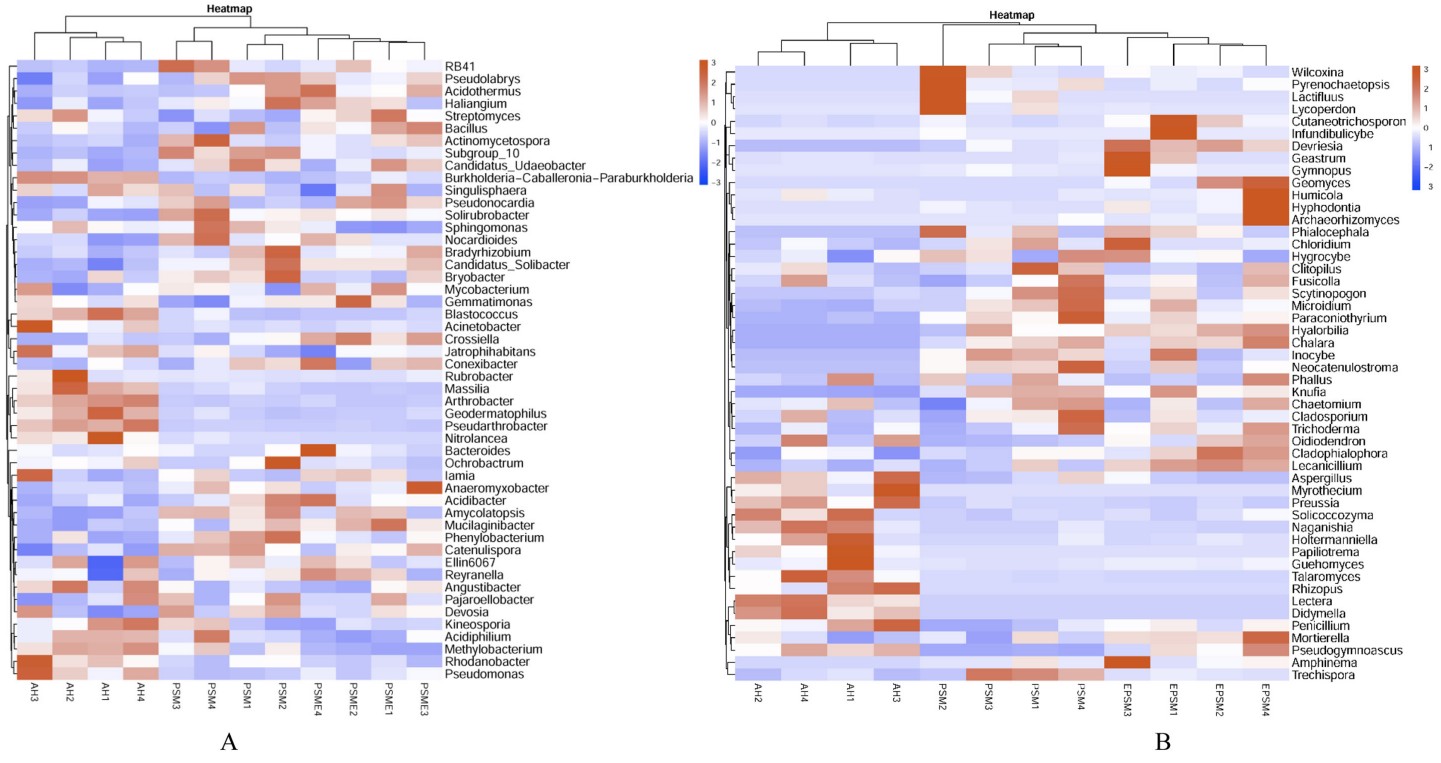

**Figure 6 Heatmap plots of soil bacterial (A) and fungal (B) genera with relative abundances of the top 50 based the bray distance.** AH, *Arachis hypogaea*; PSM, *Pinus sylvestris* var. *mongolica*; PSME, *Pinus sylvestris* var. *mongolica* with enclosure. The samples are grouped according to the similarity of each other, and the clustering results are arranged horizontally according to the clustering results. In the figure, red indicates that the genus with higher relative abundance of the corresponding sample, and blue indicates that the genus relative abundance of the corresponding sample is low.

P ($r = 0.66$, $P = 0.05$) contents (Fig. 8). Ascomycota showed negative relation to available N ($r = -0.62$, $P = 0.05$). While, Basidiomycota existed significantly positive correlation with available N ($r = 0.62$, $P = 0.05$). Mortierellomycota exhibited significantly positive correlation with total C ($r = 0.62$, $P = 0.05$), total N ($r = 0.61$, $P = 0.05$) and C/N ($r = 0.79$, $P = 0.01$), however, which showed negative relation to available P ($r = -0.74$, $P = 0.01$) (Fig. 8).

## DISCUSSION

### Effects of afforestation combined with enclosure management on soil properties

Afforestation is considered to be an effective option to sequester carbon in semi-arid regions (*Nosetto, Jobbágy & Paruelo, 2006*). The improvement of soil fertility after restoration of vegetation is complex ecological processes, which is influenced by numerous biotic and abiotic factors (*Cao et al., 2008*). Our results also concluded that afforestation with PSM had positive influences on soil quality and significantly increased soil total C and total N concentrations compared to AH (Table 2), which was consistent with previous study (*Li et al., 2012a*). Afforestation might facilitate the absorption of C through the accumulation of above-ground and underground biomass, and reduce the carbon loss

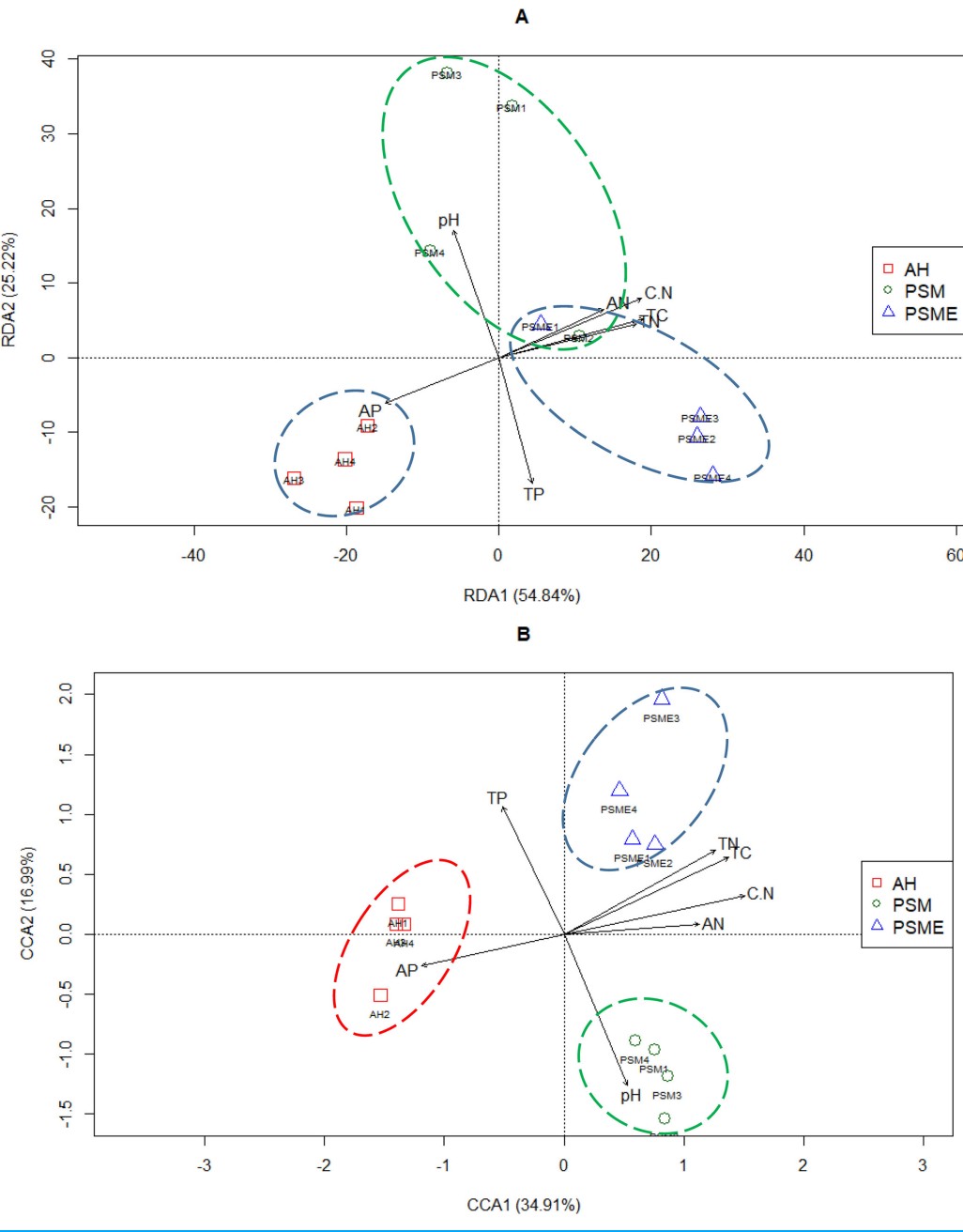

**Figure 7 Redundancy analysis (RDA) between soil environment factors and bacterial (A) and fungal OTUs (B).** TC, total C; TN, total N; TP, total P; AP, available P; AN, available N; C.N, C/N.

through retarding the decomposition of soil organic matter and soil erosion (*Nosetto, Jobbágy & Paruelo, 2006*). These results confirmed that afforestation with PSM in the Horqin Sandy Land was a positive and effective way to restore and increase soil C storage and improve soil quality in these semi-arid desertified lands (*Li et al., 2012a*). In our study, soil PH was acid in this region with highest in PSM, and lowest in PSME, which was similar to the research from forests of Uruguay demonstrating that soil pH in excluded

**Table 4 VPA of the soil environment factors on microbial community composition.**

| Soil environment factors | The relative contribution (%) | |
| --- | --- | --- |
| | Bacterial community composition | Fungal community composition |
| pH value | 2.60 | 9.20 |
| Total C | 4.39 | 7.94 |
| Total N | 4.74 | 7.81 |
| C/N | 2.50 | 8.21 |
| Total P | 7.95 | 7.33 |
| Available P | 4.64 | 6.56 |
| Available N | 3.10 | 5.56 |

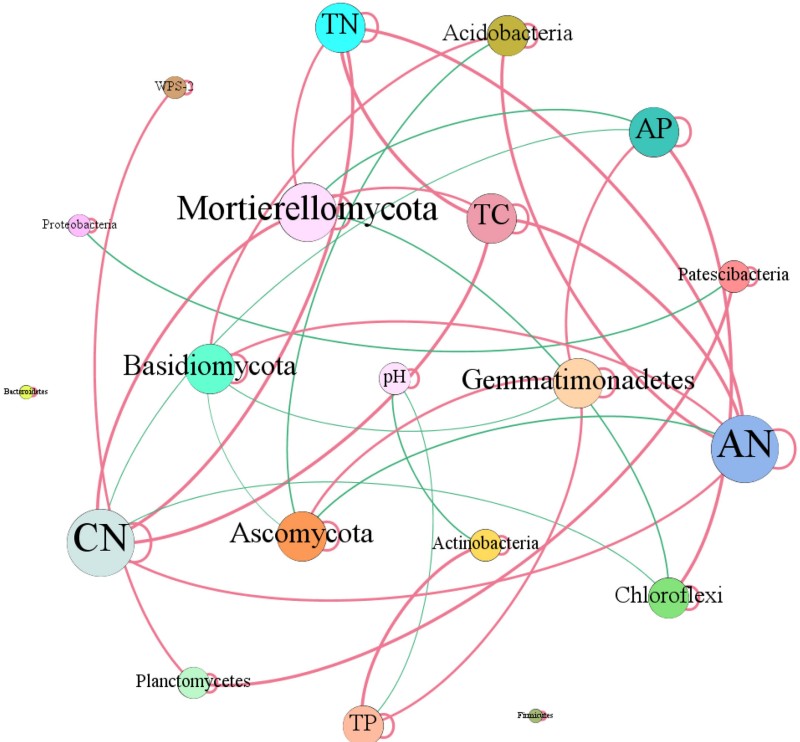

**Figure 8 Network of co-occurring of soil characteristics and dominant bacterial and fungal phyla.** A connection stands for a strong (Spearman's $r > 0.5$) and significant ($P$-value $< 0.01$) correlation. TC, total C; TN, total N; CN, C/N; TP, total P; AN, available N; AP, available P.

area was lower than in grazed area (*Etchebarne & Brazeiro, 2016*). However, previous study from Inner Mongolia grasslands finding that soil pH increased and became more neutral with the increase of enclosure time (*Ma et al., 2016*). What's more, we further found that PSME could increase soil nutrients significantly than PSM, including soil total C, total N, total P and available N contents ($P < 0.05$), which was agreement with the studies about grassland from *Xiong et al. (2016)* and *Gebregergs et al. (2019)*. The increase in total C and total N concentrations might be mainly ascribed to the undergrowth vegetation

recovery, accumulation of litter and increase in soil nutrient return in PSME (Table 1), and changes in the "quality" and "quantity" of litter lay an important role in affecting the soil nutrient (*Chastain, Currie & Townsend, 2006*). Identically, results of the impact of grazers on birch forest demonstrated that aboveground carbon stocks were higher in the long-term absence of sheep than in the continual presence of high sheep densities (*Gebregergs et al., 2019*). These results, as well as our findings, indicated that afforestation combined with enclosure management contribute to the accumulation of carbon stocks and the recovery of soil conditions in ecosystems in semi-arid regions (*Grünzweig et al., 2003*; *Laclau, 2003*; *De Souza Oliveira Filho et al., 2019*).

## Effects of afforestation combined with enclosure management on soil microbial communities

Soil microbes are an important part of soil nutrient cycling and transformation, affecting the absorption and utilization of nutrients by plants. According to the results of the soil microbial community diversity, vegetation restoration with PSM could clearly increase the soil bacterial and fungal Chao1 index and ACE index, as well as soil bacterial Shannon index than AH (Table 3), which was consistent with past observation (*Peng, Jia & Wang, 2017*). The development of plantations or enclosure management could change some soil microbes, thus affecting soil community diversity through various pathways, such as species compatibility, cooperation, and competition among microorganisms (*Yin et al., 2016*). Plantation trees cover immediately influences the understory composition and available light quantity, which affects carbon distribution and the soil microbial community (*Mitchell et al., 2012*). In addition to this, long-term fertilization also adversely affects microbial populations (*Sui et al., 2013*). Additionally, PSME had increased soil Chao1 index, ACE index of soil bacteria, and our observations agreed with the previous research that suggested fencing of degraded steppe significantly increased bacterial Chao1 index, ACE index compared with free grazing (*Zhou, Wang & Hao, 2012*). However, PSME decreased soil bacterial Simpson index and Shannon index (Table 3), which was also similar with the report by *Wang et al. (2019)* who elaborated that free grazing site significantly decreased the bacterial diversity compared to enclosure site, particularly soil Shannon index. Beyond that, extensive studies have reported that moderate castration and grazing were more conducive to soil nutrient cycling than no grazing (*Li et al., 2017b*; *Liu et al., 2016*), that likely contributed to the moderate pasture management can effectively stimulate functional microbial activity in the soil.

Soil bacterial and fungal community compositions significantly differed among AH, PSM, and PSME, and microbial community assemblage in the AH distinctly differed compared with the PSM and PSME (Figs. 2, 6 and 7). Our these results were consistent with past observation suggesting that soil microbial communities exhibited clear differences between forest (poplar plantation) and agriculture land (*Zea mays* and *Oryza sativa*) (*Cao et al., 2017*). Not only that, PSME could change soil microbial communities, especially soil fungal communities (Figs. 2 and 7), which was similar to findings from *Patra et al. (2005)* who emphasized that grazing induced the changes of soil microbial size and composition. These results, as well as our findings, elaborated that afforestation

combined with enclosed management might influence the structures of microbial communities to some extent. In addition, we found the dominant bacterial and fungal phyla among AH, PSM and PSME were nearly the same, although the relative abundances of them were disparate. The predominant bacterial phylum was Actinobacteria (Fig. 4A), which was similar with findings of *Liu et al. (2014)* and *Peng, Jia & Wang (2017)*. While, our result was dissimilar to past survey demonstrating that Proteobacteria was the most abundant phylum (*Urbanová, Šnajdr & Baldrian, 2015*). Actinobacteria is one of the main group of bacteria and play an important role in carbon cycling and organic matter turnover (*Zheng et al., 2017*), which predominate under stressful and harsh soil conditions (*Teixeira et al., 2010*). In our study, we found that Actinobacteria decreased markedly in PSM relative to AH, similar results were found in *Zheng et al. (2017)* who observed that relative abundance of Actinobacteria decreased markedly when croplands were converted to monoculture plantation. Interestingly, we further found that the relative abundances of Cyanobacteria and Chloroflexi increased in site PSM relative to PSME, which obtain energy and fix $CO_2$ via photosynthesis (*Klappenbach & Pierson, 2004*), and to a certain degree offset the reduction in plant carbon sequestration, which in turn enormously confirmed the observations of grassland (*Yao et al., 2018*). On the contrary, Nitrospirae decreased under PSM than PSME, agreement with observations of *Yao et al. (2018)* and *Lücker et al. (2010)*. At the genus level, *Crossiella*, *Jatrophihabitans*, *Conexibacter* and *Sphingomonas* were the dominant gerera (Fig. 5A), which was inconsistent with the observation demonstrating that *Seudomonas* and *Acinetobacter* were the dominant genera (*Wang et al., 2019*).

With regard to soil fungi, the dominant fungal communities were Ascomycota, Basidiomycota and Mortierellomycota (Fig. 4B). Basidiomycota and Ascomycota are the main decomposers of soil fungi, accounting for more than 90% of the total number of total fungal phyla (*Vandenkoornhuyse et al., 2002*; *Bastian et al., 2009*). To the best of our knowledge, Basidiomycota play vital roles in regulating the decomposition of low quality lignification and aromatic substrates (*Six et al., 2006*). Our results demonstrated that the relative abundances of Basidiomycota in PSM and PSME were higher than in AH, which confirmed that the relative abundance of Basidiomycota gradually accumulate with the increases of the undergrowth vegetation diversity, tree cover and litter content (*Toljander et al., 2006*). This finding explored why pine forest soils with higher C/N than agriculture land harbored a high prevalence of Basidiomycota. Interestingly, afforestation with PSM could signally increase soil ectomycorrhizal fungus, such as *Amphinema* and *Wilcoxina* (Fig. 5B). What's more, PSME site improved the relative abundance of *Amphinema*, and reduced the relative abundances of *Wilcoxina* relative to PSM (Fig. 5B). Amphinema plays an important role in plant nutrient absorption, which is known as widespread ECM species and efficient root colonizers (*Kranabetter, 2004*; *Menkis et al., 2011*; *Vaario et al., 2009*). Wilcoxina belongs to Ascomycetous ECM fungi (*Lazarević & Menkis, 2018*), which have been commonly reported in association with conifer seedlings, (*Menkis et al., 2005*). It is speculated that intensive agriculture, afforestation combined with enclosed management could greatly influence the diversity and structure of the soil microbial communities.

## The correlations between soil variations and microbial community composition

Increasing evidence indicates that environmental conditions are key factors in shaping the structures of microbial communities (*Hanson et al., 2012*). The biogeochemical processes of terrestrial ecosystems are significantly affected by afforestation (*Deng et al., 2016*), which could alter soil properties (C, N and P) and their ecological stoichiometry (*Ren et al., 2016*), thus affecting the structure and function of soil microbial communities (*Lauber et al., 2013*). In our study, soil bacterial Chao1 index and ACE index showed significantly positive correlation with total C, total N, and available N and obviously positive relationship with C/N (Fig. 3A), in line with previous observations (*Li et al., 2014*). In present study, the soil microbial community structure were affected by multiple environmental factors (Fig. 7), which was similar to other results (*Griffiths et al., 2011*; *Deng et al., 2020*). RDA plots illustrated that changes in bacterial community structure were associated soil pH, C/N, total C, total P, total N, available N and available P (Fig. 7A), which was accordance with the results from *Chao et al. (2016)* and *Santonja et al. (2018)* who reported that soil nutrients (total C, total N, total P and available P) were the main factors influencing the bacterial communities, similar observations were observed (*Zornoza et al., 2015*; *Zeng, Dong & An, 2016*; *Li et al., 2017a*). Soil organic carbon also influenced soil Actinobacteria and Acidobacteria abundances, as previous research have reported that Acidobacteria are more abundant in soil with relatively high soil organic carbon and low organic carbon soil consistently exhibits the highest abundance of Actinobacteria (*Sul et al., 2013*). These were inconsistent with our observations that the relative abundance of Actinobacteria were negatively correlated with pH and positively related with soil total P (Fig. 8). Previous study found that the relative abundances of Acidobacteria were positively correlated with soil pH (6.67–9.01) (*Yao et al., 2018*). However, no similar results were found in our study, possibly because the soil pH range in our study was relatively small (5.61–5.95) (Table 2).

In case of soil fungal community structure, CCA plot suggested that soil pH, total C, total N, C/N, total P, available N, available P were the dominant driving factors (Fig. 8), which was similar with the results obtained in the previous studies (*Lauber et al., 2008*; *Yang et al., 2014*; *Deng et al., 2019*). Basidiomycota existed significantly positive correlation with available N, similar results were obtained in Chinese pine plantations on the Loess Plateau (*Dang et al., 2017*). While, Ascomycota showed negative relation to available N, which was inconsistent with past study demonstrating that soil phosphorous is considered an important regulator of Ascomycota in the soil (*Lauber et al., 2013*). The results illustrated that afforestation combined with enclosed management potentially regulate microbial properties through shifting the soil properties.

## CONCLUSIONS

In summary, our data demonstrated that there were obviously differences in soil chemical parameters and soil microbial communities among different samples. Afforestation with PSM had positive influences on soil quality and significantly increased soil total C and total N concentrations compared to AH. Our results confirmed that the enclosure with

PSM was a positive way to restore and improve soil quality in these semi-arid desertified lands. Vegetation restoration with PSM could clearly increase the soil bacterial community diversity and fungal Chao1 index and ACE index. Additionally, enclosure of PSM could further increase soil Chao1 index, ACE index of soil bacteria. Soil total C, total N and available N concentrations in this area were the main factors affecting the soil microbial community diversity. Afforestation with PSM might remarkably shift soil microbial communities, and the microbial communities between PSM and PSME were similar but also clearly different. The effects of PSME on soil fungal communities were significantly greater than those of bacterial communities. What's more, the soil microbial community structure were affected by multiple environmental factors, and changes in soil chemical parameters induced by afforestation and enclosed management potentially regulated microbial properties. This study provides deep insights into the effects of afforestation with *Pinus sylvestris* var. *mongolica* combined enclosure management on soil microbial communities in Zhanggutai and Horqin sand ecosystems.

### Funding

This research was financially supported by the Sub-project of the National Key Research and Development Program (2017YFC050410501), the Special Fund for Forestry Scientific Research in the Public Interest (No. 201404303-05), the National Science and Technology Support Program of China (2015BAD07B30103), the Special Fund for Forest Scientific Research in the Public Welfare (201304216). The funders had no role in study design, data collection and analysis, decision to publish, or preparation of the manuscript.

### Grant Disclosures

The following grant information was disclosed by the authors:
Sub-project of the National Key Research and Development Program: 2017YFC050410501.
Forestry Scientific Research in the Public Interest: 201404303-05.
National Science and Technology Support Program of China: 2015BAD07B30103.
Forest Scientific Research in the Public Welfare: 201304216.

### Competing Interests

The authors declare that they have no competing interests.

### Author Contributions

- Jiaojiao Deng conceived and designed the experiments, performed the experiments, analyzed the data, prepared figures and/or tables, authored or reviewed drafts of the paper, and approved the final draft.
- Yongbin Zhou conceived and designed the experiments, prepared figures and/or tables, and approved the final draft.
- Wenxu Zhu conceived and designed the experiments, performed the experiments, analyzed the data, authored or reviewed drafts of the paper, and approved the final draft.

- You Yin analyzed the data, prepared figures and/or tables, authored or reviewed drafts of the paper, and approved the final draft.

## Field Study Permissions

The following information was supplied relating to field study approvals (i.e., approving body and any reference numbers):

Field work was conducted at the Zhanggutai forest farm. The director of the forest farm (Liqiang Shui) allowed us to carry out experiments in the forest farm.

## Data Availability

The raw high-throughput sequencing data are available at NCBI SRA: PRJNA562091 (bacteria) and PRJNA562096 (fungi).

## Supplemental Information

Supplemental information for this article can be found online at http://dx.doi.org/10.7717/peerj.8857#supplemental-information.

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
