# Peer review of "Effects of afforestation with Pinus sylvestris var. mongolica plantations combined with enclosure management on soil microbial community"

_PeerJ, doi:10.7717/peerj.8857_

## Round 0.1 · original submission · Major Revisions

Dear authors,

Thank you for your interest and submission to PeerJ. After reading the reviewer's comments and concerns regarding every section of the manuscript, I feel that, in its present state, it is not ready to be published here. However, However, a resubmission of the manuscript could be considered if a thorough revision is conducted and all issues raised in the review process will be addressed adequately.

Here are some specific points stressed below that should be seriously taken into account: a. comments about inappropriate or incomplete statistics; b. insufficient problem statement; c. incomplete, inaccurate, or outdated review of the literature, and specific in the discussion section the inadequate link of findings to practice.

If you choose to submit a revision, you should include the point by point response to reviews in your resubmission cover letter.

Thank you for giving us the opportunity to consider your work.

Kind regards,
Gabriela Nardoto

Reviewer 1 ·

Basic reporting

This is very clear and interesting descriptive study on soil micro biota under different treatments.
There is a clear background, but more details may be needed (see below)

Experimental design

the experimental design seems to be appropriate. See below for comments on some clarifications that may be needed.

Validity of the findings

Results are robust and clear, but some extra analyses may be needed .

Additional comments

This is very clear and interesting descriptive study on soil (micro) biota under different treatments in China. This provides very useful information of how different scenarios may drastically change soil biota. I consider that this text is very clear, but some aspects may need some clarification.

There are some typos in the text (not many, but some), so please take a look at it in detail again. For example, in the “Determination of vegetation type and soil samples collection” section, the first sentence in not clear. I suggest you change it to “We selected four 1-ha plots of Pinus sylvestris var. mongolica plantations…”. In line 180, change the comma for a dot. It should read “existed. In order”. Change the word castration after litter. I am not exactly sure what you mean there, but castration is generally use in animal ecology for something very different, so it is confusing. These are just examples; please take a closer look at the text.

Explain better why Arachis hypogaea is an ideal control. This is not clear why you used it. Is it because it is may vegetation when pines are not there? I think that in “Site information” you need to explain more about the local ecosystems (e.g. pines are native there or they are just use for forestry?).

Since you worked with pine afforestation, I think is important if you talk about mycorrhizal species. Specially ectomycorrhizal species. You mention Wilcoxina in the text, that is a very common mycorrhizal species. You do a very detailed analysis of the diferent groups of fungi and bacteria, but it would be super interesting to know the role that each have. For mycorrhizal species, this may not be too hard to do and I think can add a lot (do mycorrhizal species decrease or increase under grazing?)

All in all, I think that this is a very solid contribution, but some details may be needed to make it even more interesting and relevant to the international scientific community.

Reviewer 2 ·

Basic reporting

I was looking forward to reading this manuscript. It turned out to be a rather disappointing because of the lack of clear objectives, hypotheses and by the experimental system (peanut - a cultivated species of South American origin vs. Pinus).
The manuscript is wholly descriptive, and the findings are readily predictable (verge on the commonplace). The langue requires considerable polishing.

Experimental design

I found the experimental set-up rather strange. Why did the authors choose to compare enclosed and unenclosed Pinus plantation with a peanut field, instead of native (degraded?) vegetation?
Research questions are not well defined, or relevant / meaningful. Nor is stated how the research reported fills identified knowledge gaps.
The Methods are somewhat difficult to read to replicate field design / sampling.

Validity of the findings

As there are no well-defined original research questions the discussion and the conclusions are difficult to judge.

Additional comments

The authors have failed to establish:

(1) why the submitted work is important and timely (what lacunae in knowledge it addresses),
(2) what hypotheses or questions the reported work addresses ,
(3) how the work fills the identified gaps in knowledge and how the work overall advances our understanding.

Annotated reviews are not available for download in order to protect the identity of reviewers who chose to remain anonymous.

Reviewer 3 ·

Basic reporting

This article is well writing. The research topics are clear and the article structure conforms to the structure of scientific articles. However, the topic of this research is not very innovative; it is a common issue with several similar reports. However, relatively few related references have been cited. I suggest that this study can compare previous studies and put forward more direct hypotheses and address the differences and novelty of this study from other studies.

Experimental design

The experimental design for the manipulation is not a big problem. Nevertheless, I have some questions in statistical analysis. Since RDA is performed, I would like to see the proportion of explanation and significance of each constraint factor (the significance of each factor can be tested by type-II ANOVA). Besides, since constrained factors may be collinear between each other because various soil factors may interfere or affect each other. I recommend that you exclude collinear factors before performing RDA to avoid unnecessary weighting caused by multicollinearity.

In addition, I think it is necessary to explain why RDA and CCA are done.

Validity of the findings

no comment

Additional comments

Some other comments are listed below:
Introductions
1. What is enclosure management? How is enclosure management performed in this forest?
2. Addition to "better understand the afforestation effect combined with enclosure management", are there any clearer research questions and hypotheses of this study?

Materials and Methods:
1. RStudio is a program to help conduct R. You should use functions in packages of R to perform the analyses with the assistance of RStudio instead of using packages in RStudio for the analysis.
2. R is sensitive to the upper and lower cases, so the packages used in R should be written carefully in upper/lower case. Packages vegan and car, for example, should be written in lower case (vegan and car), instead of Vegan and Car.
3. How many samples used in this study?
4. L233-234: Write the F statistic behind each soil factors rather than put all of the F values together.

Results and Discussion
1. In Table 2, the P value should not be <0.00.
2. Detailed data (including statistical data) should be displayed in the tables, not in the main text. In the main text, you should put the summarized comparisons and statistics instead of details.
3. In the discussion, the soil microbes mediate the carbon sequestration is interpreted. The discussion about causality should be more cautious, because correlation cannot be directly equivalent to causality, it is just correlation.
4. Many paragraphs are too long, especially the discussion. I would suggest segmentation.
5. The authors believe that afforestation can increase soil carbon and nitrogen concentrations. Although the authors have data to support higher soil carbon and nitrogen concentrations in afforested land, this composition may also be that the original carbon and nitrogen concentrations of these grounds are high. Also, vegetation restoration and enclosure management were suggested to be able to increase the richness of microbial communities in soils. Are there more direct evidences that afforestation will directly increase soil carbon and nitrogen concentrations? Furthermore, is there any evidence that the increase of microbial richness is not just due to the high richness of microorganisms in these plots originally? I have read that the authors have cited several papers to support these inferences. However, if just citing various references to support these inferences rather than providing more manipulated or statistic evidences, what is the value of this research data? Therefore, I would expect to see more proper manipulated or statistic evidences, rather than just providing more references.
6. Species compatibility, cooperation, and competition among microorganisms are quite high. When the external environment changes, the change of microorganisms will not be in a single taxon, but a set or sets of functional groups. These comparisons of microbial community diversity before and after the afforestation or enclosure management should take into account interactions between species, at least some discussion should be provided.
7. I think that the beta-diversity should also be compared directly, not just comparing the difference between alpha-diversities of different communities.

---

## Round 0.2 · accepted · Accept

Dear Dr. Zhu,

after thoroughly reading the new version of the manuscript, it is clearly an improved version. The reviewers and I really appreciate the work you and your co-authors have done answering the reviewers' comments. It is a pleasure for me to write to you informing that your manuscript has been accepted for publication in PeerJ.

All the best,

Gabriela Nardoto

Reviewer 1 ·

Basic reporting

the authors have responded to all my concerns and now the manuscript is stronger and clearer. Thanks for your work on this!

Experimental design

the authors have responded to all my concerns and now the manuscript is stronger and clearer. Thanks for your work on this!

Validity of the findings

the authors have responded to all my concerns and now the manuscript is stronger and clearer. Thanks for your work on this!

Additional comments

the authors have responded to all my concerns and now the manuscript is stronger and clearer. Thanks for your work on this!